# Circulating Tumor DNA Early Kinetics Predict Response of Metastatic Melanoma to Anti-PD1 Immunotherapy: Validation Study

**DOI:** 10.3390/cancers13081826

**Published:** 2021-04-11

**Authors:** Guillaume Herbreteau, Audrey Vallée, Anne-Chantal Knol, Sandrine Théoleyre, Gaëlle Quéreux, Emilie Varey, Amir Khammari, Brigitte Dréno, Marc G. Denis

**Affiliations:** 1Laboratoire de Biochimie et Plateforme de Génétique Moléculaire des Cancers, Centre Hospitalier Universitaire (CHU) Nantes, 44093 Nantes, France; guillaume.herbreteau@chu-nantes.fr (G.H.); audrey.vallee@chu-nantes.fr (A.V.); sandrine.charpentier@chu-nantes.fr (S.T.); 2Centre de Recherche en Cancérologie et Immunologie (CRCINA), Institut National de la Santé et de la Recherche Médicale (INSERM) U1232, 44007 Nantes, France; anne-chantal.knol@univ-nantes.fr (A.-C.K.); gaelle.quereux@chu-nantes.fr (G.Q.); emilie.varey@chu-nantes.fr (E.V.); amir.khammari@chu-nantes.fr (A.K.); brigitte.dreno@atlanmed.fr (B.D.); 3Service de Dermatologie, Centre Hospitalier Universitaire (CHU) Nantes, 44000 Nantes, France; 4Centre d′Investigation Clinique (CIC) 1413, Institut National de la Santé et de la Recherche Médicale (INSERM), Centre Hospitalier Universitaire (CHU) Nantes, 44021 Nantes, France

**Keywords:** immunotherapy, anti-PD1, cell-free DNA, circulating tumor DNA, melanoma, metastatic melanoma, digital PCR, follow-up, monitoring, criteria

## Abstract

**Simple Summary:**

The early detection of primary resistance to anti-PD1 immunotherapies remains a major challenge in the management of metastatic melanoma. In a previous study, we suggested that early monitoring of circulating tumor DNA (ctDNA) using well-defined evaluation criteria allows the identification of primary resistance to anti-PD1 immunotherapies as early as the second week of treatment. We used the same criteria to analyze first-month ctDNA kinetics in a validation cohort. We confirmed that an initial “biological progression” (i.e., a significant increase in ctDNA levels) was an early predictor of a complete lack of clinical benefit under anti-PD1, both in the validation cohort and by pooling the validation and derivation cohorts. Moreover, ctDNA detection at first-line treatment initiation was an independent prognostic factor for overall survival and progression-free survival. The results confirm that early quantitative ctDNA monitoring can detect primary resistance of metastatic melanoma to anti-PD1 immunotherapies.

**Abstract:**

The ability of early (first weeks of treatment) ctDNA kinetics to identify primary resistance to anti-PD1 immunotherapies was evaluated with a validation cohort of 49 patients treated with anti-PD1 for metastatic BRAF or NRAS-mutated melanoma, alone and pooled with the 53 patients from a previously described derivation cohort. BRAF or NRAS mutations were quantified on plasma DNA by digital PCR at baseline and after two or four weeks of treatment. ctDNA kinetics were interpreted according to pre-established biological response criteria. A biological progression (bP, i.e., a significant increase in ctDNA levels) at week two or week four was associated with a lack of benefit from anti-PD1 (4-month PFS = 0%; 1-year OS = 13%; *n* = 12/102). Patients without initial bP had significantly better PFS and OS (4-month PFS = 78%; 1-year OS = 73%; *n* = 26/102), as did patients whose ctDNA kinetics were not evaluable, due to low/undetectable baseline ctDNA (4-month PFS = 80%; 1-year OS = 81%; *n* = 64/102). ctDNA detection at first-line anti-PD1 initiation was an independent prognostic factor for OS and PFS in multivariate analysis. Overall, early ctDNA quantitative monitoring may allow the detection of primary resistances of metastatic melanoma to anti-PD1 immunotherapies.

## 1. Introduction

Anti-Programmed-Death receptor 1 (anti-PD1) antibodies, alone or in combination with anti-CTLA-4 antibodies, have become a standard treatment for patients with advanced melanoma, increasing both progression-free survival (PFS) and overall survival (OS) in metastatic melanoma patients compared to chemotherapy and CTLA-4 inhibitors alone [1,2,3,4,5]. Nevertheless, approximately 60% of patients do not respond, and their identification remains a major challenge [1,2,3].

This issue has generated significant interest in the development of tumor biomarkers for monitoring the therapeutic response of metastatic cutaneous melanoma to checkpoint inhibitor immunotherapies. Circulating tumor DNA (ctDNA) is quantitatively associated with tumor burden [6], and several studies have shown that (i) a high baseline ctDNA is associated with poor OS in melanoma patients, independent of treatment, LDH, and tumor stage [7,8,9,10,11]; and (ii) quantitative changes in ctDNA are correlated with melanoma response to targeted therapies [12,13,14,15,16] and immunotherapies [13,16,17,18,19,20,21,22], with a decrease in ctDNA observed during therapeutic response and an increase in ctDNA associated with progressive disease.

A major limitation to the use of ctDNA monitoring is the lack of standardized evaluation criteria to interpret its kinetics in order to reliably identify primary resistance and allow for therapeutic changes. Some studies have assessed the detectability of ctDNA during follow-up [10,11,14,23,24]; however, the notion of detectability depends on the sensitivity of the technique used and overlooks quantitative changes in cDNA above the detection limit of the method. ctDNA quantitative monitoring may be more relevant [25,26], but it is necessary to define the levels of variation at which ctDNA changes have clinical significance.

Considering that a change in ctDNA can be clinically significant only if it exceeds the imprecision of the assay, we previously defined quantitative biological response and progression criteria for the interpretation of digital PCR-measured ctDNA kinetics (see Material and Methods, Table 1) [20]. Digital PCR (dPCR) is an absolute ctDNA quantification method that evaluates the precision of quantification at each measurement. Using this feature, we defined the biological response (bR) as a statistically significant decrease in plasma ctDNA compared to its baseline level considering the accuracy of dPCR. Similarly, biological progression (bP) was defined as a statistically significant increase in ctDNA compared to its nadir. In this derivation study, the absence of bR as early as week two was associated with an absence of benefit of anti-PD1 immunotherapy (ORR = 0%; 4-month PFS = 0%).

This validation study was undertaken to confirm that early ctDNA kinetics, interpreted using well-defined evaluation criteria, could predict responses to anti-PD1 immunotherapy alone or in combination with anti-CTLA-4 in an independent prospective cohort of metastatic melanoma patients.

## 2. Materials and methods

### 2.1. Study Design

Patients who started anti-PD1 immunotherapy treatment for a BRAF or NRAS-mutated stage IV or unresectable stage III metastatic cutaneous melanoma were included in the derivation cohort between January 2014 and March 2017, and in the validation cohort between March 2017 and January 2019. Treatment and response assessment has been previously described [20]. EDTA plasma samples were collected for each patient before the initiation of treatment and at the 2nd week of treatment, or at the 4th week of treatment if no sample was collected at week 2. ctDNA was extracted and quantified in *BRAF* or *NRAS*-mutated copies/mL of plasma by dPCR, using the methodology previously described [20]. If the mutation was undetectable, the concentration of ctDNA was considered to be below the detection limit of dPCR (i.e., 4 mutated copies/mL of plasma with our extraction and analysis conditions).

Quantification accuracy was assessed for each measurement. Briefly, dPCR partitions the sample into several thousand microcompartments and isolates each copy of the gene of interest in an individual microcompartment. After allele-specific end-point TaqMan PCR, the number of mutated copies is extrapolated from the proportion of mutation-positive microcompartments, using Poisson’s law. The repeatability of the measurement can be estimated from the standard error of this proportion.

### 2.2. Interpretation of ctDNA Kinetics

Digital PCR can estimate the accuracy of its quantification for each measurement. Thanks to this feature, ctDNA concentrations between two monitoring points can be compared using a simple statistical test (comparison of the proportion of dPCR mutation-positive microcompartments, one-sided Z-test; if the mutation was undetectable at one monitoring point, the proportion was considered to be below 1/total number of microcompartments). ctDNA kinetics between baseline and the first follow-up point were evaluated according to our previously established criteria [20] (Table 1).

### 2.3. Statistical Analysis

Patient characteristics were compared using Fisher and Mann–Whitney non-parametric tests. Survival probabilities were estimated using the Kaplan–Meier method and compared using the log-rank test. A Cox proportional hazards model was used to assess the prognostic value of baseline ctDNA detection across the whole cohort and within subgroups. Firth’s penalized likelihood was used for subgroup analysis, to allow for survival analysis despite low number of events (progression or death) in some subgroups. Clinical and biological variables associated with survival in the subgroup analysis were included in a multivariate analysis. Statistical analyses of this study were performed using the XLSTAT and R software programs.

## 3. Results

### 3.1. Validation Cohort

The validation study included 49 patients with stage IV or unresectable stage III BRAF or NRAS-mutated metastatic cutaneous melanoma. Patient characteristics at treatment initiation are presented in Table 2: 44 patients were treated with nivolumab alone and 5 were treated with a nivolumab-ipilimumab combination. BRAF codon 600 mutations were found in 16 patients: 5 were treated with first-line nivolumab, 10 were treated with nivolumab after a targeted therapy, and 1 patient was treated with a nivolumab-ipilimumab combination after a targeted therapy. NRAS mutations were found in 33 patients: 23 were treated with first-line nivolumab alone and 4 were treated with a first-line nivolumab-ipilimumab combination, 3 were treated with nivolumab after treatment with ipilimumab, and 3 patients resumed treatment with nivolumab after a relapse following the discontinuation of an initial anti-PD1 immunotherapy.

The median follow-up duration was 15.0 months (min–max = 0.7–30.3 months). Thirty patients (61%) were alive at the time of analysis, and 26 patients (53%) still had a response to anti-PD1 immunotherapy; 4-month PFS, 1-year PFS, and 1-year OS were 75%, 49%, and 79%, respectively.

ctDNA was detectable at baseline in 19 patients (40%). The detection of baseline ctDNA was associated with stage, gender, the presence of abdominal or bone metastases, and LDH activity (Table 2).

### 3.2. Early ctDNA Monitoring in Validation Cohort

For the 49 patients in the validation cohort, ctDNA concentration was quantified early at week 2, or at week 4 when no sample was collected at week 2 (*n* = 32 and 17 respectively; median = 14.0 days; Q1–Q3 = 13.8–28.0 days).

At the first follow-up point, a bP was observed in 4 patients, a bS in 5 patients, and a bR in 8 patients. Thirty-two patients (*n* = 32) had a non-evaluable biological response, including 30 patients whose baseline ctDNA was undetectable (none showed a significant increase in ctDNA at the first follow-up point) and 2 patients whose basal ctDNA concentration was too low to identify a bR, despite an undetectable ctDNA at the first follow-up point.

An initial bP was associated with a 4-month PFS = 0% (median PFS = 52.5 days; Figure 1). The detection of a bR at the first follow-up point was associated with a 4-month PFS = 88% and a 1-year PFS = 58% (median PFS not reached). bS defined an intermediate group, with a 4-month PFS = 80% and a 1-year PFS = 40% (median PFS = 268 days).

Patients with an initial bP had a significantly lower PFS than bR patients (HR = 13.61; 95%CI = [2.35–140.91]; *p* = 0.003; Figure 1) and bS patients (HR = 5.90; 95%CI = [1.04–60.86]; *p* = 0.045). The difference in PFS between bR and bS patients was not significant.

These results were comparable to those observed at the first follow-up point in our first derivation study: an initial bP during follow-up was associated with a lack of response to anti-PD1 immunotherapies. Given their comparability, the two cohorts were pooled for subsequent analyses.

### 3.3. Pooled Analysis

The derivation and validation cohorts together included 102 patients with BRAF or NRAS-mutated stage IV or unresectable stage III metastatic cutaneous melanoma. Baseline ctDNA was detectable for 44 patients (43%) and was associated with stage, gender, number of metastases, the presence of abdominal or bone metastases, and baseline LDH activity (Table 3). More women had a less advanced disease than men (stage IIIc: 42.1% vs. 20.0%; *p* = 0.020). They presented fewer metastatic sites (mean: 3.5 vs. 4.4, respectively; *p* = 0.019), lymph node metastases (presence of lymph node metastases: 68.4% vs. 91.1%; *p* = 0.007), and bone metastases (12.3% vs. 28.9%; *p* = 0.046) than men.

In this pooled analysis, the median follow-up duration was 10.8 months (min–max = 0.7–42.0 months). Sixty-six patients (*n* = 66; 65%) were alive at the endpoint of each cohort, and 57 patients (56%) were still responding to anti-PD1 immunotherapy; 4-month PFS, 1-year PFS, and 1-year OS were 72%, 48%, and 71%, respectively.

### 3.4. Early ctDNA Monitoring in Pooled Analysis

At the first follow-up point, of the 102 patients in the pooled analysis, a bP was observed in 12 patients, a bS in 8 patients, and a bR in 18 patients. Due to low or undetectable baseline ctDNA, the biological response was non-evaluable (NE) for 64 patients.

Similar to the validation cohort, bP at the first follow-up point was associated with 4-month PFS = 0% (median PFS = 84 days; Figure 2A). The detection of an initial bR was associated with a 4-month PFS = 83% and a 1-year PFS = 62% (median PFS not reached). Patients with an initial bR had a significantly longer PFS (HR = 12.7; IC95% = [3.8–53.3]; *p* < 0.0001; Figure 2A) and OS (HR = 7.5; IC95% = [2.6–24.1]; *p* = 0.0002; Figure 2B) than bP patients. bS patients defined an intermediate group with a 4-month PFS = 63% and a 1-year PFS = 42% (median PFS = 268 days).

### 3.5. Prognostic Value of Baseline ctDNA Detection

Baseline ctDNA detection was associated with a poor OS in univariate analysis (HR = 2.62; 95%CI = [1.33–5.18]; *p* = 0.006). There was no significant association between baseline ctDNA detection and PFS (HR = 1.73; 95%CI = [0.96–3.11]; *p* = 0.067).

Considering the therapeutic line, ctDNA detection at the initiation of first-line immunotherapy was clearly associated with both PFS (HR = 3.70; 95%CI = [1.54–8.93]; *p* = 0.004; Appendix A) and OS (HR = 7.14; 95%CI = [2.36–21.67]; *p* = 0.001; Appendix A), while there was no association between baseline ctDNA detection and PFS or OS at subsequent treatment lines (Appendix A–D).

Fifty-eight patients (*n* = 58) were treated with first-line immunotherapy (see Appendix A for patient characteristics). In a multivariate analysis, ctDNA detection at the initiation of first-line immunotherapy was still associated with poor OS, after adjustment for stage, mutated gene, treatment, LDH subgroup (> or ≤2 × upper limit of normal), and the number and nature of metastases (HR = 10.52; 95%CI = [1.83–60.55]; *p* = 0.008; Appendix A). The prognostic value of baseline ctDNA detection also remained significant in multivariate analysis, considering LDH as a continuous variable (HR = 8.92; 95%CI = [1.10–72.32]; *p* = 0.040).

Baseline ctDNA detection before first-line immunotherapy was associated with poor PFS after adjustment for age, gender, stage, mutated gene, treatment, LDH subgroup (> or ≤2 × upper limit of normal), tumor thickness and ulceration, and the number and nature of metastases (HR = 5.52; 95%CI = [1.22–24.90]; *p* = 0.026; Appendix A). The association between PFS and baseline ctDNA detection, however, did not remain significant in multivariate analysis, considering LDH as a continuous variable (HR = 4.83; 95%CI = [0.56–41.48]; *p* = 0.151).

### 3.6. Biological Follow-Up Model

Overall, the majority of patients (64/102; 63%) had a non-evaluable biological response (NE; i.e., no bP and a low or undetectable baseline ctDNA). These patients had a good prognosis in OS and few of them showed primary resistance to immunotherapy (4-month PFS = 80%; 1-year OS = 81%; Figure 3).

Among patients with an evaluable biological response, the 26 patients with an initial bR or bS at the first follow-up point also had a favorable OS and a low primary resistance rate to immunotherapy (4-month PFS = 78%; 1-year OS = 73%; Figure 3). These patients did not significantly differ from NE patients in terms of PFS or OS.

On the other hand, the 12 patients with an initial bP at the first follow-up point showed a total lack of benefit from immunotherapy and a poor prognosis in OS (ORR = 0%; 4-month PFS = 0%; 1-year OS = 13%; Figure 3). These bP patients had a significantly lower PFS than bR or bS patients (HR = 9.2; 95%CI = [2.9–28.6]; *p =* 0.0001) or NE patients (HR = 10.2; 95%CI = [4.1–25.3]; *p* < 0.0001). Similarly, bP patients had a significantly lower OS than bR or bS patients (HR = 3.7; 95%CI = [1.5–9.1]; *p =* 0.005) or NE patients (HR = 7.8; 95%CI = [3.4–18.2]; *p* < 0.0001).

These 12 patients with an initial bP represented 46% of patients who had a progressive disease during the first 4 months of treatment, and bP preceded the radiological detection of clinical progression by an average of 55.8 days (median = 49 days; min–max = 13–98 days).

## 4. Discussion

This study confirms that a biological progression (i.e., a significant increase in ctDNA concentration compared to its nadir, considering the inaccuracy of the measurement) at week 2 or week 4 of an anti-PD1 immunotherapy, alone or in combination with an anti-CTLA-4, allows the early and highly specific detection of patients with primary resistance to treatment.

The early detection of primary resistance to anti-PD1 immunotherapies may represent a major advance in the management of metastatic cutaneous melanoma. Indeed, there is currently no sufficiently reliable predictive biomarker of the efficacy of immunotherapies to guide the therapeutic strategy. PD-L1 tumor expression failed to show sufficient specificity to exclude a possible therapeutic response associated with anti-PD1 immunotherapies, with ORRs ranging from 13% to 41% in PD-L1-negative patients treated with nivolumab alone, and even 55% in PD-L1-negative patients treated with the nivolumab-ipilimumab combination [27]. The distribution and density of lymphocytes infiltrating the tumor showed better specificity in predicting the efficacy of anti-PD1 immunotherapies, with two studies reporting negative predictive values of 100% [28,29]. The clinical utility of this biomarker, however, remains severely limited by its inter- and intra-tumor heterogeneity and may require repeated invasive biopsies. Failing prior selection of patients likely to respond to anti-PD1 immunotherapies, the use of ctDNA as a biomarker for minimally invasive monitoring could enable the early detection of non-responders in the initial phase of treatment. Additionally, we showed in this study that patients with initial bP had a poor prognosis in OS, and the clinical benefit associated with rapid therapeutic adaptation could be major in them.

The evaluation criteria for interpreting longitudinal variations in ctDNA remain unclear. For instance, Lee et al. proposed qualitative criteria based on ctDNA detectability in dPCR: patients with a detectable ctDNA up to the 12th week of treatment had a worse ORR and PFS compared to patients whose ctDNA was initially undetectable or became undetectable before week 12 (3-month ORR = 6% vs. 74%; 3-month PFS = 28% vs. 83%, respectively) [19]. These qualitative criteria could not, however, exclude a possible benefit associated with anti-PD1 immunotherapies, and the 12-week delay required to identify non-responders limited their clinical relevance. Assessing quantitative variations in ctDNA without defining their significance could also lead to misinterpretations. For instance, in the results recently reported by Váraljai et al. [22], an increase in ctDNA levels at week 12 (regardless of significance) did not exclude a possible efficacy from immunotherapies (5-month PFS = 30%; 1-year PFS = 15%).

The interpretation of ctDNA kinetics during monitoring is complex, however, as ctDNA quantification in dPCR can be inaccurate, especially when the number of mutated copies is low, with a coefficient of variation (CV) that can be close to 100% as the method approaches its detection limit. Consequently, it seems essential to assess the significance of ctDNA longitudinal variations in relation to measurement inaccuracy before they can be given clinical meaning. The significance of ctDNA variations defining bR and bP, however, cannot be defined by fixed variation thresholds (i.e., % change from a reference). Indeed, the use of such fixed thresholds would require the accuracy of dPCR to be constant regardless of ctDNA concentration, but the accuracy of dPCR is concentration-dependent: the lower the concentration of mutated copies detected, the higher the CV of dPCR [30,31]. Thus, a fixed variation threshold is likely to be lower than the CV at low ctDNA concentrations, and a simple measurement inaccuracy would be interpreted as a bP or bR, while the same threshold will be much higher than the CV at high concentrations and will limit the detection of small ctDNA variations.

The accuracy of ctDNA quantification by dPCR can be assessed at each measurement. With this feature, it is possible to determine whether a difference in ctDNA concentration between two monitoring points is significant using a simple statistical test (comparison of the proportion of dPCR mutation-positive microcompartments between the two points, one-sided Z-test). Our method of interpretation was recently used by Wood-Bouwens et al. in a small study of 6 patients with metastatic cancers (breast cancer, colorectal cancer, cholangiocarcinoma, melanoma), with a good concordance with tumor evolution [32].

ctDNA monitoring requires knowledge of a specific somatic mutation, previously identified in the tumor tissue, to quantify the fraction of circulating DNA of tumor origin. Cutaneous melanoma is a particularly suitable model since BRAF or NRAS hotspot mutations are found in 60% to 70% of patients and are already routinely investigated for any metastatic cutaneous melanoma [33,34]. Nevertheless, in the absence of a previously identified somatic mutation, ctDNA cannot be used for therapeutic monitoring, which is a first limitation of this biomarker. Other genes frequently mutated in metastatic cutaneous melanoma (KIT, TERT, etc.) could also be used to identify and quantify ctDNA in a larger proportion of patients. The use of ctDNA assays capable of screening a large panel of genes, however, seems necessary to be able to carry out biological monitoring of all patients, and to extrapolate this biological monitoring model to other tumor types, for which mutation hotspots are less frequent. To this end, ctDNA quantification by Next-Generation Sequencing (NGS) could be an interesting alternative, but mutated DNA quantification by NGS is only relative (in % of mutated alleles), and therefore also subject to quantitative variations of non-tumoral circulating DNA.

The need for a detectable baseline ctDNA to allow biological monitoring is a second limitation of this biomarker, particularly in our study, since ctDNA was only detectable in 43% of patients in the derivation cohort and 40% of patients in the validation cohort, i.e., a lower sensitivity than the data reported in the literature concerning dPCR methods (with sensitivities of around 70% to 80% [10,18,22,35,36,37,38]). This limitation must, however, be put into perspective: we also showed in this study that undetectable baseline ctDNA was informative, since it was a good prognostic factor in OS and PFS at the initiation of first-line immunotherapy, independently of other known prognostic factors for metastatic cutaneous melanoma. These results are consistent with those observed by several authors [12,13,14,15]. We recently showed in another study that the lack of detection of ctDNA by qPCR Cobas (Roche diagnostics) at the initiation of first-line treatment was an independent factor of good prognosis, regardless of treatment (immunotherapy or targeted therapy) in BRAF or NRAS-mutated melanoma patients [16].

Overall, ctDNA monitoring appears feasible, rapid, and minimally invasive. The analysis of ctDNA before the initiation of first-line treatment would allow the rapid detection of BRAF or NRAS mutations for theranostic purposes and could clarify the prognosis of the disease. Subsequently, for patients treated with anti-PD1 immunotherapy alone or in combination with anti-CTLA-4, a second ctDNA analysis at week 2 or week 4 could assess the biological response, provide early identification of some primary resistances to immunotherapy, and allow rapid therapeutic adaptation. Prospective clinical trials comparing radiological and biological monitoring of ctDNA with conventional follow-up are needed to evaluate the clinical benefit associated with this monitoring model.

## 5. Conclusions

In conclusion, the early quantitative ctDNA monitoring can detect primary resistance of metastatic melanoma to anti-PD1 immunotherapies. It now seems necessary to standardize analytical methods and interpretation criteria for ctDNA kinetics and to validate its clinical value through prospective trials.

## Figures and Tables

**Figure 1 cancers-13-01826-f001:**
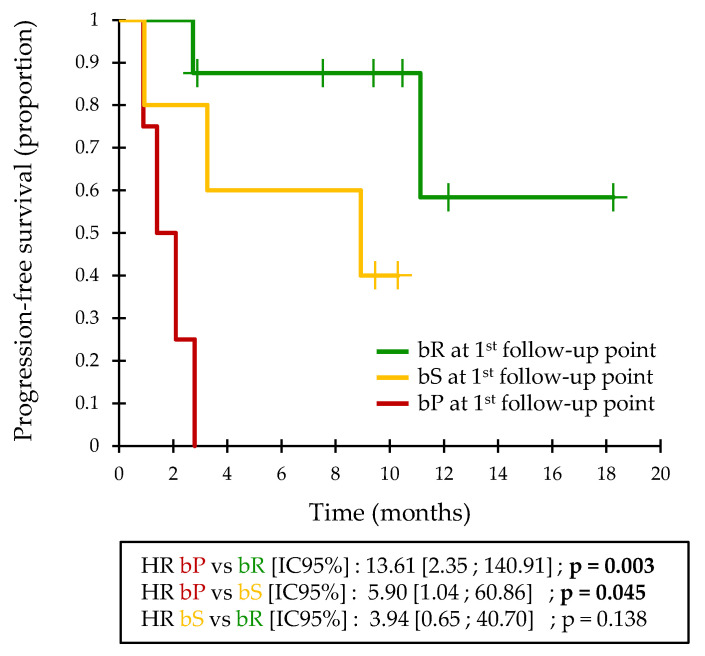
Kaplan–Meier estimate of the PFS in the validation cohort, based on the biological response assessment at the first follow-up point.

**Figure 2 cancers-13-01826-f002:**
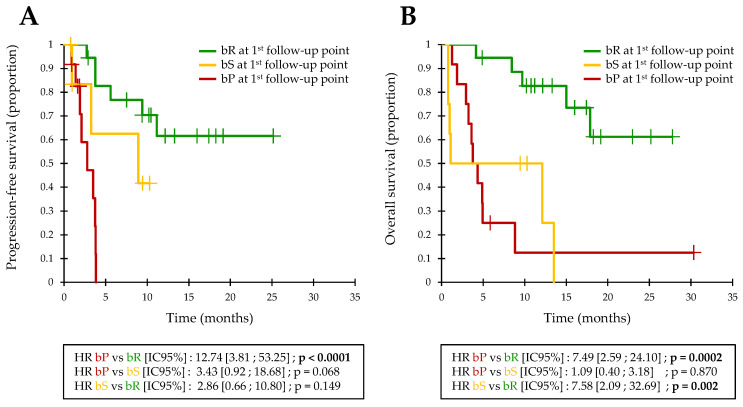
Kaplan-Meier estimate of the PFS (**A**) and OS (**B**) in pooled analysis, based on the biological response assessment at the first follow-up point.

**Figure 3 cancers-13-01826-f003:**
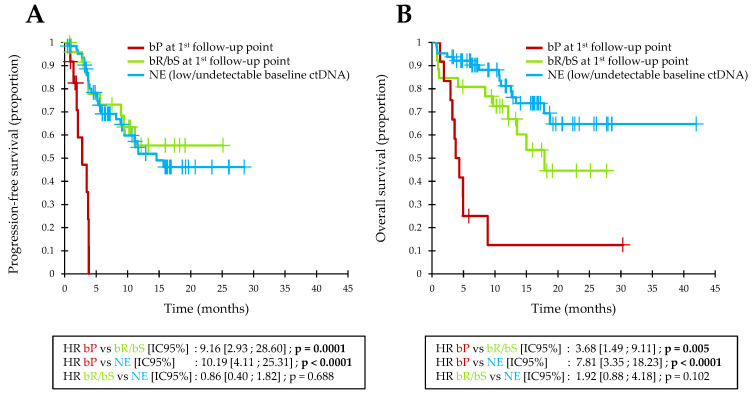
Kaplan–Meier estimates of PFS (**A**) and OS (**B**) in pooled analysis, depending on baseline ctDNA and biological response assessment at the first follow-up point.

**Table 1 cancers-13-01826-t001:** Biological evaluation criteria.

Evaluation Criteria	Definition
biological Response (bR)	Statistically significant decrease in ctDNA concentration compared to baseline, considering the accuracy of the measurement at both points (one-sided Z-test, α = 2.5%)
biological Progression (bP)	Statistically significant increase in ctDNA concentration compared to nadir, considering the accuracy of the measurement at both points (one-sided Z-test, α = 2.5%)
biological Stability (bS)	- No bR and bP criteria - Baseline ctDNA at a sufficiently high concentration to identify a bR if ctDNA became undetectable during follow-up
Non-evaluable biological response (NE)	- No bP criteria - Undetectable baseline ctDNA, or at a concentration too low to identify a bR if ctDNA became undetectable during follow-up

**Table 2 cancers-13-01826-t002:** Patient characteristics in the validation cohort.

	Total		Undetectable Baseline ctDNA	Detectable Baseline ctDNA	*p*
***n***	49		30	19	-
**Age** m (Q1–Q3)	63.6 (54.1–74.8)		65.3 (54.0–77.1)	60.9 (57.1–70.0)	0.662
Tumor thickness m (Q1–Q3)	3.3 (1.4–3.8)		2.9 (1.3–3.3)	3.8 (1.5–6.0)	0.573
Number of metastases m (Q1–Q3)	3.0 (2.0–3.0)		2.7 (2.0–3.0)	3.6 (2.0–4.0)	0.066
Baseline LDH IU/L; m (Q1–Q3)	378.3 (195.9–387.7)		221.2 (195.9–227.9)	548.5 (275.3–468.3)	0.026
Gender	M	16	6 (37%)	10 (63%)	0.028
	F	33	24 (73%)	9 (27%)	
Stage	III	22	18 (82%)	4 (18%)	0.009
	IV	27	12 (44%)	15 (56%)	
Ulceration	Yes	20	9 (45%)	11 (55%)	0.128
	No	19	13 (68%)	6 (32%)	
Presence of lymph node metastasis	Yes	32	17 (53%)	15 (47%)	0.135
	No	17	13 (76%)	4 (24%)	
Presence of cutaneous metastasis	Yes	28	20 (71%)	8 (29%)	0.139
	No	21	10 (48%)	11 (52%)	
Presence of pulmonary metastasis	Yes	11	6 (55%)	5 (45%)	0.729
	No	38	24 (63%)	14 (37%)	
Presence of cerebral metastasis	Yes	10	5 (50%)	5 (50%)	0.480
	No	39	25 (64%)	14 (36%)	
Presence of abdominal metastasis	Yes	14	3 (21%)	11 (79%)	0.001
	No	35	27 (77%)	8 (23%)	
Presence of bone metastasis	Yes	9	2 (22%)	7 (78%)	0.019
	No	40	28 (70%)	12 (30%)	
Mutated gene	NRAS	33 ^a^	20 (61%)	13 (39%)	1.00
	BRAF	16 ^b^	10 (63%)	6 (37%)	
Baseline LDH	>426 IU/L (2 × ULN)	4	0	4 (100%)	0.027
	≤426 IU/L (2 × ULN)	21	13 (62%)	8 (38%)	
	Undetermined	24	17 (71%)	7 (29%)	
Treatment	Nivolumab monotherapy	44	29 (66%)	15 (34%)	0.067
	Nivolumab + Ipilimumab	5	1 (20%)	4 (80%)	
Therapeutic line	First line	32	19 (59%)	13 (41%)	0.767
	≥second line	17	11 (65%)	6 (35%)	

^a^ p.Q61R (c.182A > G), *n* = 16; p.Q61K (c.181C > A), *n* = 12; p.Q61L (c.182A > T), *n* = 4; p.Q61H (c.183A > T), *n* = 1. ^b^ p.V600E (c.1799T > A), *n* = 14; p.V600K (c.1798_1799delinsAA), *n* = 2. ULN: upper limit of normal.

**Table 3 cancers-13-01826-t003:** Patient characteristics in pooled analysis.

	Total		Undetectable Baseline ctDNA	Detectable Baseline ctDNA	*p*
*n* (n_derivation_ + n_validation_)	102 (49 + 53)		58	44	-
Age m (Q1–Q3)	63 (54–74.6)		62.9 (52.4–75.4)	63.1 (58.3–73.5)	0.545
Tumor thickness m (Q1–Q3)	3.2 (1.5–3.9)		2.9 (1.4–3.5)	3.6 (1.6–5.0)	0.284
Number of metastases m (Q1–Q3)	3.9 (2.0–5.0)		3.7 (2.0–4.0)	4.2 (2.0–5.3)	0.011
Baseline LDH IU/L; m (Q1–Q3)	293.2 (167.7–290.2)		194.8 (160.1–223.1)	397.3 (184.5–456.6)	0.008
Gender	M	45	18 (40%)	27 (60%)	0.003
	F	57	40 (70%)	17 (30%)	
Stage	III	33	25 (76%)	8 (24%)	0.010
	IV	69	33 (48%)	36 (52%)	
Ulceration	Yes	35	17 (49%)	18 (51%)	0.345
	No	40	26 (65%)	14 (35%)	
Presence of lymph node metastasis	Yes	80	42 (53%)	38 (48%)	0.144
	No	22	16 (73%)	6 (27%)	
Presence of cutaneous metastasis	Yes	59	33 (56%)	26 (44%)	0.842
	No	43	25 (58%)	18 (42%)	
Presence of pulmonary metastasis	Yes	32	15 (47%)	17 (53%)	0.199
	No	70	43 (61%)	27 (39%)	
Presence of cerebral metastasis	Yes	22	13 (59%)	9 (41%)	1.00
	No	80	45 (56%)	35 (44%)	
Presence of abdominal metastasis	Yes	34	13 (38%)	21 (62%)	0.011
	No	68	45 (66%)	23 (34%)	
Presence of bone metastasis	Yes	20	8 (36%)	14 (64%)	0.011
	No	82	52 (63%)	30 (37%)	
Mutated gene	NRAS	62 ^a^	36 (58%)	26 (42%)	0.839
	BRAF	40 ^b^	22 (55%)	18 (45%)	
Baseline LDH	>426 IU/L (2 × ULN)	9	0	9 (100%)	0.001
	≤426 IU/L (2 × ULN)	61	36 (59%)	25 (41%)	
	Undetermined	32	22 (69%)	10 (31%)	
Treatment	Nivolumab monotherapy	93	55 (59%)	38 (41%)	0.169
	Nivolumab + Ipilimumab	9	30 (83%)	6 (17%)	
Therapeutic line	First line	58	33 (57%)	25 (43%)	1.000
	≥second line	44	25 (57%)	19 (43%)	

^a^ p.Q61R (c.182A > G), *n* = 33; p.Q61K (c.181C > A), *n* = 22; p.Q61L (c.182A > T), *n* = 5; p.Q61H (c.183A > T), *n* = 1. ^b^ p.V600E (c.1799T > A), *n* = 35; p.V600K (c.1798_1799delinsAA), *n* = 5. ULN: upper limit of normal.

## Data Availability

The data presented in this study are available on request from the corresponding author.

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
