# Peer review of "Circulating Tumor DNA Early Kinetics Predict Response of Metastatic Melanoma to Anti-PD1 Immunotherapy: Validation Study"

_cancers, 2021, doi:10.3390/cancers13081826_

Round 1

Reviewer 1 Report

The development of markers for immunotherapy efficiency is of high importance. Analysis of the level of circulating tumor DNA and cancer-specific mutations in liquid biopsy represents a good instrument for tracking tumor progression and early detection of therapy resistance. To date, there are a lot of studies that showed and validated the significance of BRAF, NRAS, and other gene mutations in circulating tumor DNA in the prediction of immunotherapy response in melanoma patients: Schreuer et al., J Transl Med 2016; Wong et al., JCO Precision Oncology 2017; Haselmann et al., Clin Chem 2018; Seremet et al., J Transl Med 2019; Keller et al., Acta Derm Venereol 2019; Braune et al., JCO Precision Oncology 2020; Kozak et al., Tumori Journal 2020; Syeda et al., Lancer Oncol 2021, etc. Some of these studies have been conducted in larger clinical cohorts. Considering these and many other works, I don’t see what new the reviewed manuscript provides. I would like to advise the authors to emphasize the novelty and importance of their study in comparison with other works. Besides, the validation group size seems to be not enough because 95% coincidence intervals are too wide.

Reviewer 2 Report

I am persuaded by the conclusions of your study, in essence you have found that a validation cohort investigating whether ctDNA is  prognostic biomarker for advanced melanoma when teated using anti-PD1 immunotherapies. There are no issues with the methods, or presentation which are clear and sound.

There are several things I wonder about that you may wish to address.

Firstly, to what extent if any is ctDNA and independent prognostic marker for patients subjected to immunotherapy. Further, is ctDNA a prognostic marker for advanced melanoma per se. Ideally, analysis in this regard should be included or at least a relevant discussion. Is the association with immunotherapy outcome merely incidental or is it only relevant when immunotherapy is used? What about LDH, it's value as a prognostic marker has been recognised, how do ctDNA and LDH compare?

This is also relevant to the approach taken with the two cohorts, which essentially represent different intervals on the same overall collection. The key point seems to be validation and yet the data has been combined. What new insight did combining the data provide, as compared to simply comparing the two sets, if they were analysed independently?

The greater proportion of cases were not evaluable, based on the detectability of ctDNA. A discussion of the quantitative limits of detection in this regard would be useful, compared to the value of detecting a significant difference between the dPCR of the early and later samples per patient.

A feature of the results is the association with gender. This should be discussed. Is the difference because of biological factors or is it a question of presentation? Are their clinical differences between the male and female patients and does this have any significance for the analysis?

Additional discussion of the power of other liquid, prognostic biomarkers for melanoma is merited so that the relative value of ctDNA is clear.

Reviewer 3 Report

This paper is very well written, showing impressiv data. The limitations are illustrated and the statistical methods are well described. The publication is very much recommended.

Author Response

We wish to thank Reviewer #3 for his/her very kind comments on our work.

Round 2

Reviewer 1 Report

The authors tried to address my criticism and provided thee detailed description of the current approaches for the analysis of ctDNA and the advantages of the approach presented in the manuscript. However, as indicated in the rebuttal letter, there are other studies that assessed changes of ctDNA over time, and the authors also used this approach. Also, I don't still see the novelty of the reviewed work. The authors say "We obtained important results compared to the literature, because we were able to identify early and specifically a group of patients with primary resistance to anti-PD1."; however, these results were obtained in the previous study (Herbreteau et al., Oncotarget 2018). It seems that the reviewed manuscript is only the validation of the results presented earlier and has no significant novelty. If it is so, I would recommend the authors to add the phrase "validation study" or something like in the manuscript title. If I am mistaken, the authors should indicate the novelty of their results clearly. 

Also, the following comments should be addressed:

  • Limit of detection of the ctDNA assay and how it was calculated;
  • Specificity/sensitivity of the ctDNA analysis for prediction of anti-PD1 response and prognosis of patient survival;
  • How often BRAF- and NRAS-positive ctDNA was detected in patients both with BRAF and NRAS mutations;
  • BRAF codon 600 and NRAS mutation types observed in the studied patients. Have mutations been detected in other BRAF codons?
  • Why the authors decided to analyze ctDNA at the points “week 2” and “week 4”? Does it have any biological or clinical significance?

Round 3

Reviewer 1 Report

The authors have answered all my comments.